# Non-line-of-sight snapshots and background mapping with an active corner camera

Sheila Seidel [1,2,5], Hoover Rueda-Chacón [1,3,5], Iris Cusini [4], Federica Villa [4], Franco Zappa [4], Christopher Yu[2] & Vivek K Goyal [1] ✉

The ability to form reconstructions beyond line-of-sight view could be transformative in a variety of fields, including search and rescue, autonomous vehicle navigation, and reconnaissance. Most existing active non-line-of-sight (NLOS) imaging methods use data collection steps in which a pulsed laser is directed at several points on a relay surface, one at a time. The prevailing approaches include raster scanning of a rectangular grid on a vertical wall opposite the volume of interest to generate a collection of confocal measurements. These and a recent method that uses a horizontal relay surface are inherently limited by the need for laser scanning. Methods that avoid laser scanning to operate in a snapshot mode are limited to treating the hidden scene of interest as one or two point targets. In this work, based on more complete optical response modeling yet still without multiple illumination positions, we demonstrate accurate reconstructions of foreground objects while also introducing the capability of mapping the stationary scenery behind moving objects. The ability to count, localize, and characterize the sizes of hidden objects, combined with mapping of the stationary hidden scene, could greatly improve indoor situational awareness in a variety of applications.

The challenge of both active and passive NLOS imaging techniques is that measured light returns to the sensor after multiple diffuse bounces[1]. With each bounce, light is scattered in all directions, eliminating directional information and attenuating light by a factor proportional to the inverse-square of the path length. Particularly in the passive setting, where no illumination is introduced, occluding structures that limit possible light paths have been used to help separate light originating from different directions in the hidden scene[2–10]. Useful structures include the aperture formed by an open window[2] or the inverse pinhole[11] created when a once-present object moves between measurement frames. Unlike other occluding structures, whose shapes must be estimated or somehow known[4,5,7,8,12], vertical wall edges have a known shape and are often present when NLOS vision is desired. An edge occluder blocks light as a function of its azimuthal incident angle around the corner and, as a result, enables

computational recovery of azimuthal information about the hidden scene. This was first demonstrated in the passive setting[3,13], where 1D (in azimuthal angle) reconstructions of the hidden scene were formed from photographs of the floor adjacent to the occluding edge; 2D reconstruction was demonstrated in a controlled static environment, although the longitudinal information present in the passive measurement was found to be weak[14]. Robust longitudinal resolution with passive measurement has required a second vertical edge[3,15].

In the active setting, most of the approaches proposed to date scan a pulsed laser over a set of points on a planar Lambertian relay wall and perform time-resolved sensing with a single-photon detector to collect transient information[16–25]. To reconstruct large-scale scenes, these approaches generally require scanning a large area of the relay wall and thus a large opening into the hidden volume. To partially alleviate these weaknesses, edge-resolved transient imaging (ERTI)[26]

[1]Electrical and Computer Engineering, Boston University, 8 St. Mary's Street, Boston, MA 02215, USA. [2]Charles Stark Draper Laboratory, 555 Technology Square, Cambridge, MA 02139, USA. [3]Computer Science, Universidad Industrial de Santander, Carrera 29 Calle 7, Bucaramanga, Santander 680002, Colombia. [4]Dip. Elettronica, Informazione e Bioingegneria, Politecnico di Milano, Piazza Leonardo Da Vinci, 32, Milano I-20133, Italy. [5]These authors contributed equally: Sheila Seidel, Hoover Rueda-Chacón. ✉e-mail: v.goyal@ieee.org

combines the use of an edge occluder from passive NLOS imaging with the transient measurement abilities of active systems. ERTI scans a laser on the floor along an arc around a vertical edge, incrementally illuminating more of the hidden scene with each scan position. Differences between measurements at consecutive scan positions are processed together to reconstruct a large-scale stationary hidden scene. Like with the earlier methods, the laser scanning requirement is still a limiting constraint. An earlier work using the floor as a relay surface shortens acquisition time by using a 32 × 32 pixel SPAD array in conjunction with a stationary laser[27]. Simultaneous measurements from the 1024 pixels and reference subtraction are used to track the horizontal position of a hidden object in motion, modeled as a point reflector. Conceptually, almost all methods that achieve more than point-like reconstruction treat the hidden scene as static during a scan. A notable method for moving objects requires rigid motion, while maintaining a fixed orientation with respect to the imaging device, and a clutter-free environment[28].

In this work, we maintain ERTI's parameterized reconstruction capability and its strength of requiring only a small opening into the hidden volume while eliminating ERTI's scanning requirement and creating a new background mapping capability. We use similar hardware as in ref. [27] and also use a floor as a relay surface. As illustrated in Fig. 1A, our desire for NLOS vision is caused by an occluding wall; unlike in ref. [27], the edge of the wall is explicitly modeled and exploited to enable reconstruction of moving objects in the hidden scene. Like in passive corner-camera systems[3,13–15], we position the SPAD field of view (FOV) adjacent to the wall edge, as shown in Fig. 1A, to derive azimuthal resolution from the occluding edge. As in ref. [26], we derive longitudinal resolution from the temporal response to the pulsed laser. However, our proposed system acquires data for each frame in a single snapshot without scanning, allowing us to track hidden objects in motion. The foreground reconstructions are independent across frames, with no requirement of morphological continuity. Consider Fig. 1A and note that a moving target not only adds reflected photons to the measurement, but also reduces photons due to the shadow it casts on the stationary scene behind it. Through additional modeling of occlusion within the hidden scene itself, we use these changes to reconstruct occluded background regions for each frame (Fig. 1B). As an object moves through the hidden scene, reconstructions of occluded background regions may be accumulated to form a map of the hidden scene (Fig. 1C). In contrast to refs. [27], [29],

where $x$ and $y$ coordinates are estimated for a hidden target in motion at an assumed height, our algorithm counts hidden objects in motion and reconstructs their shape (i.e., height and width), location, and reflectivity while simultaneously mapping the stationary hidden scenery occluded by them.

## Results
### Acquisition methodology

In our setup, the measurement rate at the $n$th spatial pixel in the $k$th time bin is Poisson distributed

$$\mathbf{x}^{n,k} \sim \text{Poisson}\left(\mathbf{b}^{n,k} + \mathbf{s}_{\text{fg}}^{n,k}(\boldsymbol{\psi}_{\text{fg}}) - \mathbf{s}_{\text{oc}}^{n,k}(\boldsymbol{\psi}_{\text{fg}}, \boldsymbol{\psi}_{\text{oc}})\right), \qquad (1)$$

where $\mathbf{b} \in \mathbb{R}^{N \times K}$ is the rates due to stationary scenery, $\mathbf{s}_{\text{fg}} \in \mathbb{R}^{N \times K}$ is the response of the foreground object, and $\mathbf{s}_{\text{oc}} \in \mathbb{R}^{N \times K}$ is the response of the occluded background region, before the object enters. We assume that $\mathbf{b}$ is approximately known through a reference measurement acquired before moving objects enter the hidden scene or through other means. Conceptually, this measurement of $\mathbf{b}$ allows us to subtract the counts due to an arbitrary and unknown stationary environment; the actual computations are more statistically sound than simple subtraction. Vectors $\boldsymbol{\psi}_{\text{fg}}$ and $\boldsymbol{\psi}_{\text{oc}}$ contain parameters that describe the foreground objects and corresponding occluded background regions. We seek to recover the parameters $\boldsymbol{\psi}_{\text{fg}}$ and $\boldsymbol{\psi}_{\text{oc}}$ from the measurement $\mathbf{x}$, for each measurement frame.

As shown in Fig. 2A for a single moving object, we model moving objects and their occluded background regions each as a single vertical, planar, rectangular facet resting on the ground. We assume there are $M$ moving objects with parameters $\boldsymbol{\psi}_{\text{fg}} = \{(\boldsymbol{\theta}^m, a_{\text{fg}}^m, r_{\text{fg}}^m, h^m), \quad m = 1, \ldots, M\}$. Marked in Fig. 2A, $a_{\text{fg}}^m$ is the albedo, $r_{\text{fg}}^m$ is range, and $h^m$ is height of the $m$th object. Angles $\boldsymbol{\theta}^m = (\theta_{\min}^m, \theta_{\max}^m)$ are the minimum and maximum polar angles of the foreground facet, measured around the occluding edge in the plane of the floor. The $m$th occluded region is described by range $r_{\text{oc}}^m$ and albedo $a_{\text{oc}}^m$ parameters $\boldsymbol{\psi}_{\text{oc}} = \{(a_{\text{oc}}^m, r_{\text{oc}}^m), \quad m = 1, \ldots, M\}$. The height of the occluded region is not a separate parameter; it depends upon its range $r_{\text{oc}}$ and the corresponding moving object's range $r_{\text{fg}}$ and height $h$. When parameters $\boldsymbol{\psi}_{\text{fg}}$ and $\boldsymbol{\psi}_{\text{oc}}$ have been estimated for a sequence of measurement frames, each processed separately, the vertical lines running through the centers of estimated occluded background

## A. Use scenario

occluded background

moving object

laser spot

SPAD FOV

path of motion

## B. Single-frame reconstruction

☐ reconstructed surfaces

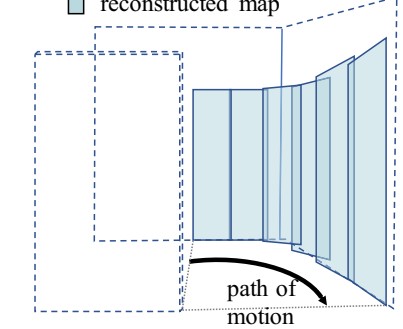

## C. Accumulating frames to map the hidden scene

☐ reconstructed map

path of motion

**Fig. 1 | Sketch of an active corner camera use case with background mapping.**
**A** The imaging equipment is on the near side of the gray occluding wall, close to the wall, and line-of-sight view ends at the extension of this wall. A pulsed laser pointed at the floor illuminates the hidden scene while a SPAD camera adjacent to the occluding wall measures the temporal response of returning light. An initial reference measurement is acquired to characterize the response of the stationary scene. When the moving object enters, the new measurement includes added photon counts due to the object and reduced photon counts at more distant ranges due to the occluded background region behind it. **B** Using these changes, we reconstruct foreground objects in motion as well as the occluded background regions behind them. **C** By accumulating frames as an object moves through the hidden scene, we form a map of the stationary background of the hidden scene.

**A**. Estimated parameters

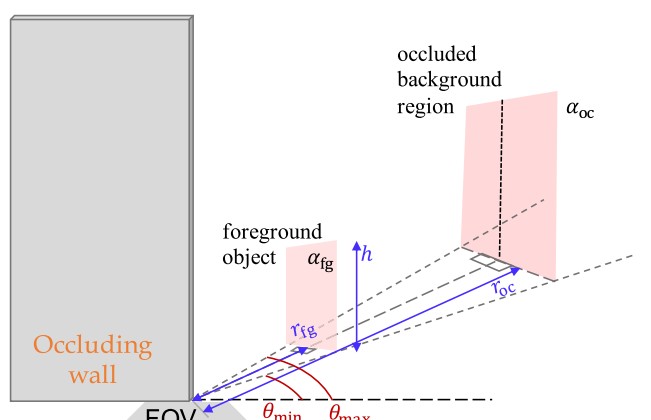

**B**. Reconstructing a map from a sequence of frames

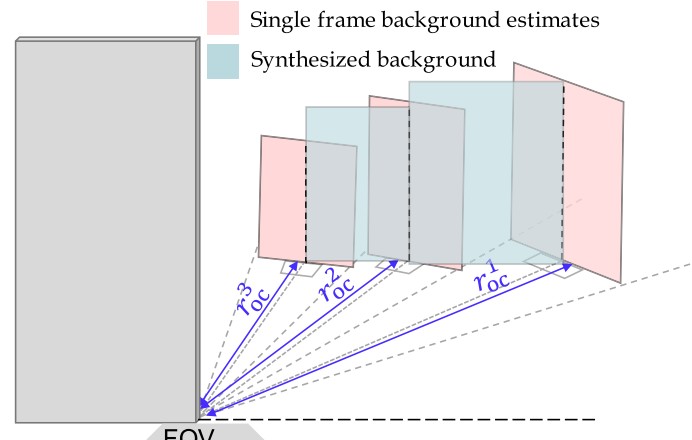

**Fig. 2 | Foreground and background facet parametrization.** Our facet-based model describes moving objects and the occluded background regions behind them as edge-facing, rectangular, planar facets characterized by the parameters shown in (**A**). When parameters have been estimated for a sequence of frames,

estimates are post-processed together to form a map reconstruction (**B**). The vertical lines going through the centers of estimated occluded regions (light red) are connected to form the map reconstruction (blue).

regions (light red) are joined by planar facets (blue) to form a contiguous map of the background, as shown in Fig. 2B.

## Fast computation of light transport

A method to quickly compute the rates due to a planar rectangular facet resting on the ground (i.e., $\mathbf{s}_{fg}(\boldsymbol{\psi}_{fg})$ and $\mathbf{s}_{oc}(\boldsymbol{\psi}_{fg}, \boldsymbol{\psi}_{oc})$) is a key part of our inversion algorithm. Take $\mathbf{p}_l$ to be the position of the laser illumination and $\mathbf{p}_f$ to be a point on the floor in the area of the $n$th camera pixel $\mathcal{P}_n$. The flux during the $k$th time bin at the $n$th camera pixel due to hidden surface $\mathcal{S}$ is

$$
\mathbf{s}^{n,k} = \int_{(k-1)\Delta_t}^{k\Delta_t} \int_{\mathcal{P}_n} \int_{\mathcal{S}} v(\mathbf{p}_s, \mathbf{p}_f) a(\mathbf{p}_s) \frac{G(\mathbf{p}_s, \mathbf{p}_l, \mathbf{p}_f)}{\|\mathbf{p}_l - \mathbf{p}_s\|^2 \|\mathbf{p}_f - \mathbf{p}_s\|^2}
$$
$$
w\left(t - t_0 - \frac{\|\mathbf{p}_l - \mathbf{p}_s\| + \|\mathbf{p}_f - \mathbf{p}_s\|}{c}\right) d\mathbf{p}_s \, d\mathbf{p}_f \, dt, \tag{2}
$$

where $a(\mathbf{p}_s)$ is the surface albedo at point $\mathbf{p}_s$, $w(\cdot)$ is the pulsed illumination waveform, $\Delta_t$ is the duration of a time bin, $t_0$ is the time the pulse hits the laser spot, and $c$ is the speed of light. The factor $G(\cdot, \cdot, \cdot)$ is the Lambertian bidirectional reflectance distribution function (BRDF) and is the product of foreshortening terms (i.e., the cosine of the angle between the direction of incident light and the surface normal), as described in Supplementary Note 1. The factor $v(\mathbf{p}_s, \mathbf{p}_f)$ is the *visibility function* that describes the occlusion provided by the occluding edge between hidden scene point $\mathbf{p}_s$ and SPAD FOV point $\mathbf{p}_f$. As shown in the bird's eye view of Fig. 3, point $\mathbf{p}_f$ is located at angle $\gamma$ measured from the occluding wall in the plane of the floor. Point $\mathbf{p}_s$ is at azimuthal angle $\alpha$, in the plane of the floor, measured around the corner from the the boundary between hidden and visible sides of the wall. Thus, point $\mathbf{p}_f$ is only visible to $\mathbf{p}_s$ if $\gamma \geq \alpha$:

$$
v(\mathbf{p}_s, \mathbf{p}_f) = \begin{cases} 1, & \text{if } \gamma \geq \alpha \\ 0, & \text{otherwise.} \end{cases} \tag{3}
$$

The yellow region in the SPAD FOV is the collection of all points $\mathbf{p}_f$ not occluded from point $\mathbf{p}_s$ by the wall, where $v(\mathbf{p}_s, \mathbf{p}_f) = 1$. In the green region, light from $\mathbf{p}_s$ is blocked by the wall and $v(\mathbf{p}_s, \mathbf{p}_f) = 0$. This fan-like pattern is the *penumbra* exploited by the passive corner camera[3,13–15].

In some previous works, computation time is reduced by making a confocal approximation[22,26] (i.e., assuming the laser and detector are co-located). Under this assumption, the set of points $\mathbf{p}_s$ in the scene that correspond to equal round-trip travel time from $\mathbf{p}_l$, to $\mathbf{p}_s$, and back

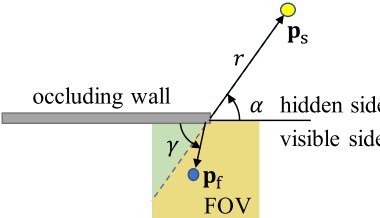

**Fig. 3 | A bird's eye view of the vertical edge occluder.** The edge blocks light from scene point $\mathbf{p}_s$ as a function of its azimuthal angle $\alpha$, measured around the corner. A point $\mathbf{p}_f$ in the SPAD FOV at angle $\gamma$ is illuminated by $\mathbf{p}_s$ if $\gamma \geq \alpha$.

to $\mathbf{p}_f$, lie on a sphere. In contrast, as in ref. 27, we seek to exploit the spatial diversity of our sensor array and thus require a more general ellipsoidal model that arises when $\mathbf{p}_l$ and $\mathbf{p}_f$ are not co-located. When $\mathcal{S}$ is a vertical, rectangular, planar facet, the intersection of a given round trip travel time (the ellipsoid) and the plane containing our facet is an ellipse. Using a method from ref. 30, we write that ellipse in translational form, enabling us to perform the integration in (2) in polar coordinates. This method, described further in Supplementary Note 1, allows us to compute $\mathbf{s}_{fg}(\boldsymbol{\psi}_{fg})$ and $\mathbf{s}_{oc}(\boldsymbol{\psi}_{fg}, \boldsymbol{\psi}_{oc})$ quickly enough to implement our inversion algorithm.

## Reconstruction approach

Before estimating parameters $\boldsymbol{\psi}_{fg}$ and $\boldsymbol{\psi}_{oc}$ for a given frame, we estimate the number of moving objects $M$. The passive corner camera processing of ref. 13 is applied to the temporally integrated difference measurement (e.g., Fig. 4B) to produce a 1D reconstruction of change in the hidden scene as a function of azimuthal angle $\alpha$. The intervals where this 1D reconstruction is above some threshold are counted to determine $M$. Parameters $\boldsymbol{\psi}_{fg}$ and $\boldsymbol{\psi}_{oc}$ are then estimated from time-resolved measurement $\mathbf{x}$ using a maximum likelihood estimate (MLE) constrained over broad, realistic ranges of $\boldsymbol{\psi}_{fg}$ and $\boldsymbol{\psi}_{oc}$. To approximate the constrained MLE, the Metropolis-Hastings algorithm is applied in two stages: first to estimate foreground parameters $\boldsymbol{\psi}_{fg}$, assuming no occlusion of the background, and second to estimate the parameters of the occluded background region $\boldsymbol{\psi}_{oc}$, assuming $\boldsymbol{\psi}_{fg} = \hat{\boldsymbol{\psi}}_{fg}$. Further details about our procedure for estimating $N, \boldsymbol{\psi}_{fg}$, and $\boldsymbol{\psi}_{oc}$ are included in Supplementary Note 2.

Since our method is not dependent on a confocal approximation, it is not important for $\mathbf{p}_l$ to be close to the base of the vertical edge

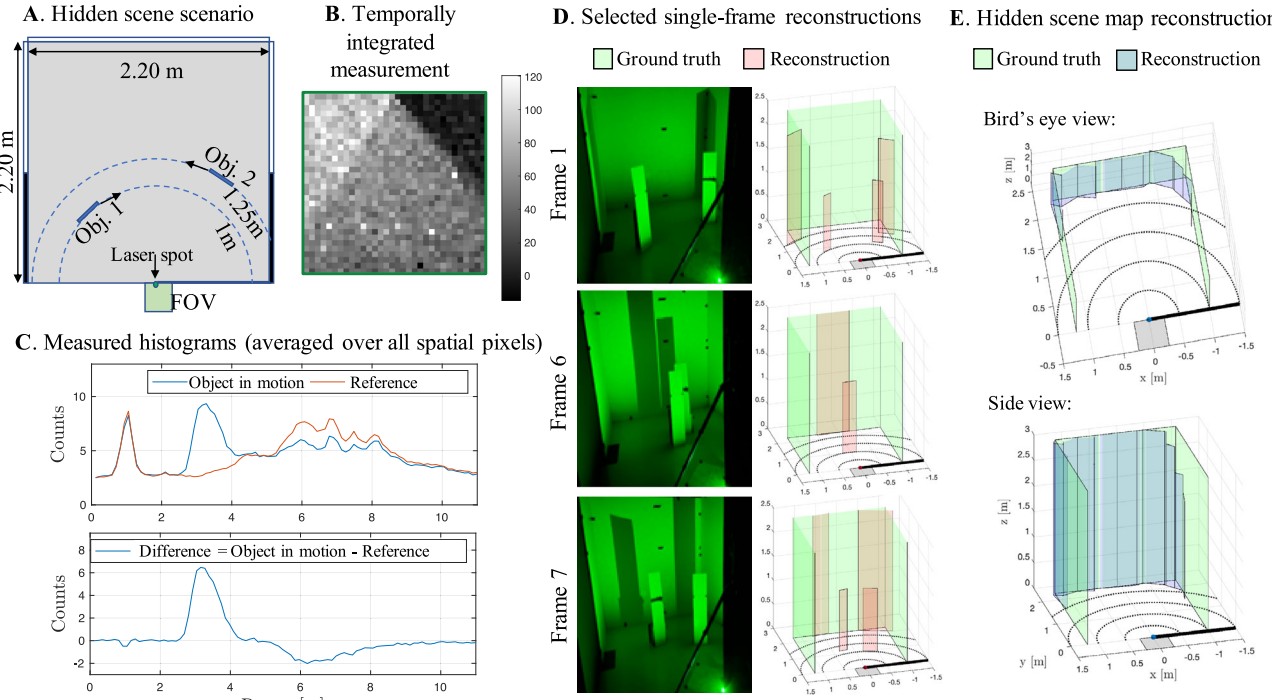

**Fig. 4 | Processing example. A** Sample measurements and reconstruction results for a scenario where two objects move through the hidden scene. Temporally integrated measurements in (**B**) show the penumbra pattern, with a distinct shadow due to each of the two hidden objects. Spatially averaged measurements for the stationary scene (red) and one motion frame (blue) are shown on the top axis of (**C**), with their difference shown below. The peak near 3 m is due to the moving objects; the dip near 6 m is due to the background occlusion. Selected single-frame reconstructions are shown in (**D**). Two views of the map reconstructions, accumulated over 8 frames, are shown in (**E**).

occluder. The imaged volume is determined by where light can reach from $\mathbf{p}_l$, as discussed in Supplementary Note 3. When neglecting the thickness of the occluding wall for simplicity, placing $\mathbf{p}_l$ at the base of the vertical edge allows the laser to illuminate the entire volume on the opposite side of the occluding wall. The placement of the SPAD FOV near the base of the vertical edge occluder is more fundamental. It enables the passive corner camera processing to be effective and greatly impacts object localization performance (ref. 31, Sect. 5.5).

**Experimental reconstructions**

In Fig. 4, we show reconstruction results for eight measurement frames acquired as two hidden objects move along arcs toward and then past each other as shown in Fig. 4A. In this demonstration, the *integration time* (i.e., the total time over which the camera collects meaningful data) used for each new frame was 0.4 s. Integration time for the reference measurement was 30 s. The *acquisition time* (i.e., the total time required to collect, accumulate, and transfer data) was longer; see Supplementary Note 3. Measurements averaged spatially over all pixels are shown in Fig. 4C. The top plot shows the stationary scene measurement (red) with the measurement acquired after objects have moved into the scene in Frame 1 (blue); their difference is shown on the axis below. A peak in the difference around 3 meters is due to the additional photon counts introduced by the two moving objects; a dip at 6 meters is due to their occluded background regions. Although it is impossible to separate the contributions from each target in this spatially integrated view of the data, the vertical edge occluder casts two distinct shadows in the temporally integrated measurement shown in Fig. 4B. Our processing exploits spatiotemporal structure of the data that is not apparent from the projections in Fig. 4B and C.

Single-frame reconstruction results are shown for three different frames in Fig. 4D. In Frames 1 and 7, two targets are resolved with accurate heights, widths, and ranges. The reconstructed occluded background regions are placed accurately in range. In Frame 6, the closer target passes in front of the more distant one, and the single reconstructed target is placed at the range of the front-most object. Two views of the reconstructed maps (blue), accumulated over all eight measurement frames, are shown in Fig. 4E to closely match the true wall locations (green).

In Fig. 5, we demonstrate that our reconstruction algorithm works with dimmer moving objects as well as with objects that do not match our rectangular, planar facet model. Single-frame reconstruction results are shown for the white facet, a darker gray facet, a mannequin, and a staircase shaped object. In all four cases, the reconstructed foreground object is correctly placed in range. Although our model does not allow us to reconstruct the varying height profile of the stairs, we correctly reconstruct it to be wider and more to the right than the other hidden objects. In Supplementary Note 4, we demonstrate that our algorithm works under a wide range of conditions, including different hidden object locations, frame lengths, and lighting conditions.

## Discussion

In this work, we present an active NLOS method to accurately reconstruct both objects in motion and a map of stationary hidden scenery behind them. This innovation is made possible through careful modeling of occlusion due to the vertical edge and within the hidden scene itself. The algorithm presented in ref. 27 attempts only to identify a single occupied point in the hidden scene, making detailed modeling of the scene response unnecessary. In this work, we also make no assumptions about light returning from the visible scene, allowing arbitrary visible scenery to be placed at the same ranges as the hidden objects of interest. This is true in ref. 26 as well, however in their setup, with the single-element SPAD fixed in position and a very small laser scan radius, the contribution to the measurement from the visible side may be assumed constant across all measurements. In our configuration, the SPAD array has a non-negligible spatial extent resulting in a visible-side contribution that varies across the measurements. Our use

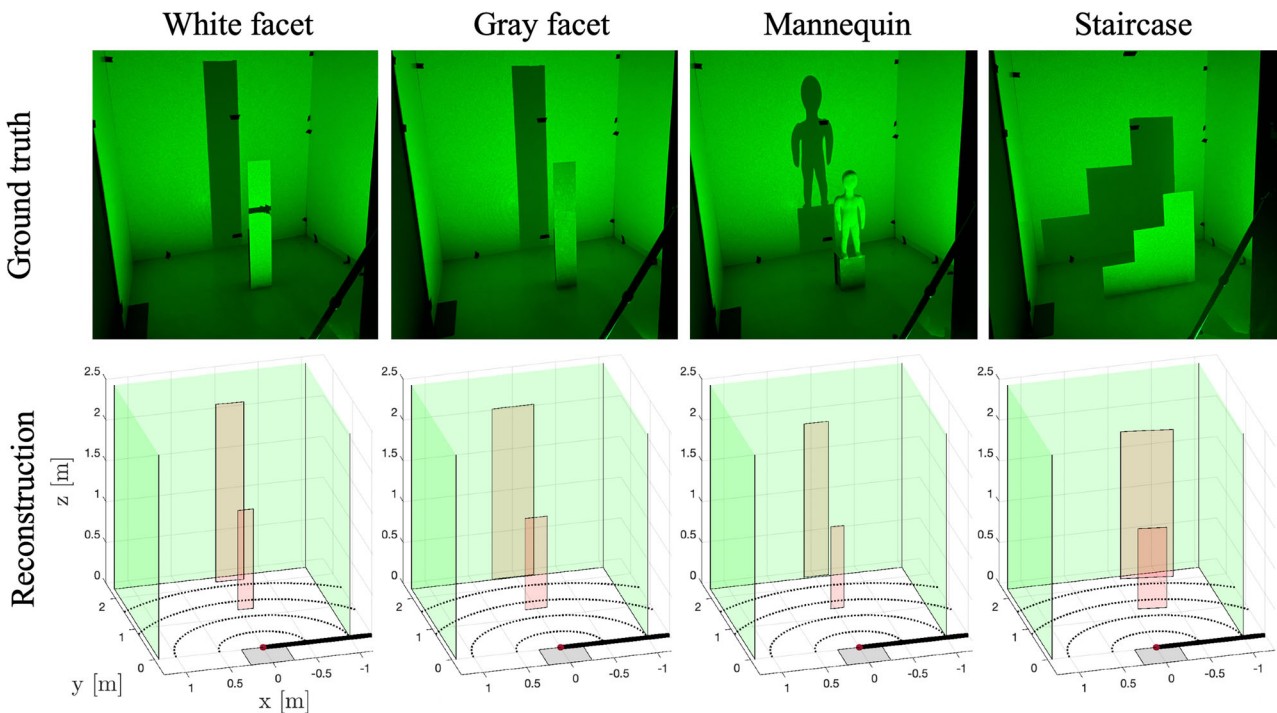

**Fig. 5 | Single-frame reconstruction results.** Four hidden objects are used: a white target, a less reflective gray target, a non-planar mannequin, and a non-rectangular staircase. In all cases, our model allows us to accurately locate both the object and the stationary scene in the background.

of a stationary scene measurement allows us to effectively remove the contribution due to unknown visible-side scenery; our modeling of occlusion within the hidden scene itself allows us to perform this background subtraction without losing all information about the stationary hidden scenery.

Although we have successfully demonstrated our acquisition method, various aspects of our system and algorithm could be improved upon. Our current algorithm processes each frame independently, using only broad constraints on the unknown parameters. An improved system could jointly process frames and benefit from inter-frame priors. Such priors could incorporate continuity of motion, the fact that object height, width, and albedo are unlikely to change between frames, and the fact that that walls in the hidden scene are typically smooth and continuous. In our demonstration, we use a thin occluding wall and do not model wall thickness. The thin-wall assumption is illustrated in Fig. 3, where the angle $\alpha$ is measured around the same point regardless of the location of $\mathbf{p}_s$. When the the wall has appreciable thickness, cases $\alpha \in [0, \pi/2)$ and $\alpha \in [\pi/2, \pi)$ require different modeling. One could incorporate wall thickness into the model or estimate wall thickness as an additional unknown parameter. A method might also be designed to produce higher resolution reconstructions of each moving target. Each target could be divided horizontally into several vertical segments, each with an unknown albedo and height to be estimated. This type of algorithm might better resolve the staircase object in Fig. 5. Through further analysis, it might also be possible to optimize certain parameters in our setup. For example, certain FOV sizes and positions or laser locations might produce a better balance between the different sources of information in the data.

The demonstrations in this work employed a sensor with $32 \times 32$ SPAD pixels, 390 ps timing resolution, 3.14% fill factor, and ~17 kHz frame rate, limited by the USB 2.0 link[32]. A frame length of 10 $\mu$s and a gate-on period of 800 ns yielded a duty cycle of 8%. Particularly, the spatial and temporal resolution limit the precision of the estimated facet parameters, whereas the fill factor and frame rate limit the signal-to-noise ratio for a given acquisition time and, thus, the ability to track

faster or farther objects. We expect the results reported in this manuscript will improve by orders of magnitude with new SPAD technology, as reviewed in refs. 33, 34, where some works have demonstrated up to 1 megapixel SPAD arrays[35], greater than 100 kHz frame rates[36], fill factors greater than 50%[36,37], and time resolution finer than 100 ps[36,38].

## Methods
### Setup
Illumination is provided using a 120 mW master oscillator fiber amplifier picosecond laser (PicoQuant VisUV-532) at 532 nm operating wavelength. The laser has an ~80 ps FWHM pulse width and is triggered by the SPAD with a repetition frequency of 50 MHz. The SPAD array consists of $32 \times 32$ pixels with a fill factor of 3.14%, with fully independent electronic circuitry, including a time-to-digital converter per pixel[32]. At the 532 nm laser wavelength and room temperature, the average photon detection probability is ~30% and the average dark count rate is 100 Hz. The array has a 390 ps time resolution set by its internal clock rate of 160.3 MHz. Attached to the SPAD is a lens with focal length of 50 mm, which yields a $25 \times 25$ cm field of view when placed at around 1.20 m above the floor. We set each acquisition frame length to 10 $\mu$s, with a gate-on time of 800 ns, thus yielding an 8% duty cycle. During the 800 ns gate-on time of each frame, 40 pulses (800 s * 50 MHz) illuminate the scene. The SPAD array has a theoretical frame rate of 100 kHz, set by the 10 $\mu$s readout per frame, but experimentally we observed just ~17 kHz, which was mainly limited by the USB 2.0 connection to the computer.

### Data acquisition
For our demonstrations, we set up a hidden room 2.2 m wide, 2.2 m deep and 3 m high, as shown in Fig. 4A. Assuming the coordinate system origin is at the bottom of the occluding edge, the left wall is at $x = -1.20$ m, the right wall is at $x = 1$ m, the back wall is at $y = 2.2$ m, and the ceiling is at $z = 3$ m. The walls are made of white foam board and the ceiling is black cloth. The SPAD array is positioned on the side of the wall, looking down at the occluding edge origin, allowing half of the array to be occluded. The laser is positioned so

that it shines close to the origin. To reject the strong ballistic contribution (first bounce) of light reflected from the origin, we punched a hole in the occluding wall and shined the laser through the hole. The true location of the laser spot on the floor is slightly off the origin, by 6 cm to the right side. The latter was found by cross-checking and minimizing the number of ballistic photons measured by the SPAD array. More recent SPAD arrays incorporate a fast hard gate to rapidly enable and disable the detector with few hundreds picoseconds width, which can be tuned to censor the ballistic photons[36,38].

Two test scenarios were analyzed. For the first, we used two rectangular white foam board facets of size $20 \times 110$ cm as our moving objects. For the second, we used four different targets: a white foam board facet (of size $20 \times 110$ cm), a gray foam board facet (white foam board painted with a gray diffuse spray paint), a fabric mannequin of size $30 \times 80$ cm, and a stair-like facet of size $75 \times 75$ cm. These objects were used to test our method on targets of different shape, height and albedo. All tests were conducted with the objects facing the occluding edge. Before moving objects enter the hidden room, a 30 s acquisition was collected to form an estimate of **b**, the response of the stationary scene. Then, new measurement frames were collected with moving objects fixed at discrete points along their trajectories during 0.4 s. In Supplementary Note 4, we demonstrate that these measurements can be acquired over a much shorter period of time with little effect on the reconstruction quality.

## Data availability
Data to reproduce the results of this paper are available on Zenodo: https://doi.org/10.5281/zenodo.7905475[39].

## Code availability
Code to reproduce the results of this paper, including a description of tuning parameters, is available on Zenodo: https://doi.org/10.5281/zenodo.7905475[39].

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

## Acknowledgements
This work was supported in part by the US National Science Foundation under grant number 1955219 (V.K.G.) and in part by the Draper Scholars Program (S.S.).

## Author contributions
S.S., H.R.C. and V.K.G. conceived the acquisition method. S.S. derived the light transport model and developed the reconstruction algorihm. S.S. and H.R.C. performed simulations. H.R.C. designed and performed the experiments. S.S., H.R.C., I.C., F.V., F.Z. and V.K.G. discussed the data. F.V., F.Z., C.Y. and V.K.G. supervised the research. All authors contributed to the manuscript.

## Competing interests
The authors declare no competing interests.

## Additional information

**Peer review information** : *Nature Communications* thanks Daniele Faccio and the other, anonymous, reviewer for their contribution to the peer review of this work. A peer review file is available.

