## [Peer Review File · Nature Communications]

Non-Line-of-Sight Snapshots and Background Mapping with an Active Corner CameraREVIEWER COMMENTS

Reviewer #1 (Remarks to the Author):

The authors present a combination of ideas, new and old, to provide a new approach to NLOS imaging whereby a non-scanning laser spot and non-scanning imaging system are used to reconstruct full 3D information of both a moving object and the static background.

The results are convincing, novel and I would expect to be of interest for a broad audience. I would therefore recommend publication in Nature Communications.

I do have a few comments/questions that the authors might want to consider.

1. I am assuming that the green dot indicates laser position in Fig4a. How does one get the laser to focus at this spot and yet the scene is still effectively beyond line of sight of the system? It seems that if the laser can be focused at the green spot, then one can also see directly inside the room?

Looking at the SM (photos of the setup), it seems that the laser spot is actually right at the very tip of the wall and not behind it, is that correct? Maybe figure4a needs to be slightly corrected. Either way, also looking at the SM, it seems that the camera in reality does always have direct line of sight access to most of the room and definitely LOS access to the objects. So it is not obvious that the geometry implemented here really counts as NLOS imaging. I am not trying to diminish the results - I still think that the approach qualifies as a form of "indirect imaging", in the sense that neither the laser nor the camera are looking directly at the scene. But I do think that the details of the geometry raise the question as to why a "stronger" NLOS of geometry was not used. My intuition is that the quality of the reconstructions decay rapidly as soon as the laser spot and camera field of view are shifted away from the edge of the wall (therefore also requiring the whole laser+camera system to move further to the right, behind the wall and/or be completely 100% occluded from any line of sight into the room, as desired for NLOS imaging). I imagine that this is because the wall edge plays a critical role and has to be within the line of sight of the camera/laser?

If so, I think this needs to be explained, ideally (but not necessarily) with an example of how things degrade when the edge is not within the line of sight of the camera+laser?

2. My impression from reading the main text was that motion was needed in order to reconstruct the position and shape of the object. But after reading the SM, if I understand correctly, motion of the object is only required in order to map the static background whereas the object itself can be (and is) reconstructed from each individual frame, i.e. in principle motion is not needed? I do not see anything in the model and maths that indicates that motion is needed for the object reconstruction, i.e. this could be done from just one single frame. Could the authors comment on this and clarify in the text?

Reviewer #2 (Remarks to the Author):

“Non-line-of-sight tracking and mapping with an active corner camera” present new results on the NLOS imaging with an edge-resolved transient imaging (ERTI) device.

This method records transient information from an inaccessible area around a corner by illuminating the hidden scene with laser light scattered from a surface (e.g. floor). With the help of an edge (e.g. door frame) in the scenery different areas are sampled. During the evaluation, a sector-by-sector analysis can be performed by difference formation.

The manuscript is well written, theory and experimental results are presented in detail and show an amazing quality.

However, in another publication (Nature Communications, 2020), parts of the author team have presented a very similar procedure.

Despite the fact that a slightly different type of approach was used, the procedures, scenarios and results are very similar that it is difficult to distinguish between them, giving the impression of marginal innovation.

In the present paper, I would have liked to have seen somewhat clearer use cases investigated and the system used outside of a controlled lab environment with less than ideal walls (smooth white surfaces). Furthermore, I am surprised that it is not possible to achieve a better reconstruction than the one already presented in 2020.

Although I really enjoyed reading this manuscript and there are no complaints in terms of content and form, I am afraid that the presented procedure and results do not represent the necessary innovation required for publication in such a prestigious journal as Nature Communications. Therefore, I cannot recommend the publication of the manuscript.

Maybe, it would be worth to submit this paper to Scientific Reports instead.

Response to Referees Letter NCOMMS-22-27501-T

Non-Line-of-Sight Tracking and Mapping with an Active Corner Camera

Sheila Seidel, Hoover Rueda-Chacón, Iris Cusini, Federica Villa,
Franco Zappa, Christopher Yu, and Vivek K Goyal

March 27, 2023

Response to Both Reviewers

We thank the reviewers for their thoughtful comments on our manuscript. We appreciate the time and effort that goes into critical reading and consideration of a manuscript with lengthy and thorough supplementary materials. Before responding to each comment in turn, we have a few general comments.

We construe academic publishing as collaborative with reviewers, not adversarial. In this spirit, we want to be open about the fact that we were required to completely disassemble our experimental setup by May 24, 2022, to accommodate a major renovation of our building. Thus, our avoidance of additional experimental work is not mere laziness; any additional data collection would require an effort that is not feasible considering that the two lead authors have departed from Boston University to start positions in industrial research and academia.

We are happy that one of two reviewers supports acceptance and the other makes many positive comments despite not being convinced of the “necessary innovation” to merit publication in *Nature Communications*. We believe that some of our conceptual and technical novelties were underappreciated, in part due to weaknesses in our writing (which we have attempted to alleviate) and in part due to hardware limitations.

Our results were achieved using a SPAD camera that is ten years old; we acquired it in 2013 for the project that eventually yielded [1]. The spatial (32×32) and temporal (390 ps) resolution of the array are not high by contemporary standards. Furthermore, this array has very limited gating and data transfer capabilities. We had arranged to have the use of superior hardware. However, since our purchase of the required XEM7360-K160T-3E FPGA faced a 52-week (!) supply chain delay, we had to proceed with old hardware. In light of this, we ask the reviewers to consciously separate *concepts* and *proof-of-principle demonstrations* from the specific *experimental achievement*. We are not asking for a low standard of scrutiny to be applied. Though we expect that we would all agree that the conceptual novelty of the work is the most important thing, we know that it is tempting to compare reconstruction resolution to [2], where the hardware is in some ways far superior. We are not claiming improved accuracy relative to [2], but instead several other novelties.

At a conceptual level, the novel capabilities here are two-fold. First, we believe that this work is unique in achieving more than crude point-like reconstructions without any scanning of the illumination. Second, it is the first active imaging work to show that shadowing of illumination by the foreground produces negative reconstructions that can be interpreted as a mapping of the background. We believe the only similar idea is in our own concurrently developed passive method [3].

We do not deny that in some ways, we are combining ideas from corner cameras [4] and ERTI [2], but that does not tell the whole story; there is significant creativity and technical development beyond that combination. In particular, it would not be fair to treat this work as an array-based version of ERTI. Like other scanning-based systems, ERTI treats the scene as static, whereas the present work combines dynamic and static imaging in a novel way. The present work does not require the scene to be static during

an illumination scan (i.e., there is no illumination scan), but it can exploit foreground movement to infer information about a static background.

The algorithms developed and deployed here are dependent on mathematical modeling that represents a significant advance beyond ERTI. To exploit the spatial diversity of the virtual sensor locations¹ provided by the array detector, we must have response modeling that does not use a confocal approximation. (A confocal approximation would treat all the virtual sensor locations as identical, eliminating their diversity.) This is conceptually easily stated, but algorithmically challenging to achieve without a large increase in computation time. Our fast but accurate approximate response computation is described in Section 1.2 of the Supplementary Notes. As we write this response, we realize that it can be problematic for a key contribution to not be detailed in the main paper, yet those details do not seem to fit in the main paper. Though we did not move details from Section 1.2 of the Supplementary Notes into the main paper, we did **insert a paragraph** starting at line 269 that draws attention to this contribution. (It also draws attention to flexibility in the location of the illumination; see the Response to Reviewer #1 for more details.) Our approximate computations for facet responses without confocal approximation could have many other applications.

To clarify the conceptual novelties of the work, we have slightly **modified the title** and have made **changes to the abstract paragraph** and the **first three body paragraphs**. This includes adding a reference to keyhole imaging [5] for additional context. Other modifications are described below.

In the following sections, we have introduced numbering to the reviewers' comments.

Response to Reviewer #1

1. The authors present a combination of ideas, new and old, to provide a new approach to NLOS imaging whereby a non-scanning laser spot and non-scanning imaging system are used to reconstruct full 3D information of both a moving object and the static background.

The results are convincing, novel and I would expect to be of interest for a broad audience.

Response:

We thank you for the positive assessment and appreciate the time and effort you put into reviewing our work. We agree that the "non-scanning/non-scanning" combination is an important distinction of this work. Additionally, we hope that the reconstruction of the room shape from the effect of occlusion by foreground objects is recognized as a valuable novelty.

2. I am assuming that the green dot indicates laser position in Fig4a. How does one get the laser to focus at this spot and yet the scene is still effectively beyond line of sight of the system? It seems that if the laser can be focused at the green spot, then one can also see directly inside the room? Looking at the SM (photos of the setup), it seems that the laser spot is actually right at the very tip of the wall and not behind it, is that correct? Maybe figure4a needs to be slightly corrected. Either way, also looking at the SM, it seems that the camera in reality does always have direct line of sight access to most of the room and definitely LOS access to the objects. So it is not obvious that the geometry implemented here really counts as NLOS imaging. I am not trying to diminish the results - I still think that the approach qualifies as a form of "indirect imaging", in the sense that neither the laser nor the camera are looking directly at the scene. But I do think that the details of the geometry raise the question as to why a "stronger" NLOS of geometry was not used. My intuition is that the quality of the reconstructions decay rapidly as soon as the laser spot and camera field of view are shifted away from the edge of the wall (therefore also requiring the whole laser+camera system to move farther to the right, behind the wall and/or be completely 100% occluded from any line of sight into the room, as desired for NLOS imaging). I imagine that this is because the wall edge plays a critical role and has to be within the line of sight of the camera/laser? If so, I think this needs to be explained, ideally

¹Many active NLOS imaging papers use *virtual sensor location* to refer to a patch on the relay surface that is the FOV of the sensor. Similarly, *virtual source location* refers to a patch on the relay surface where the illumination is focused.

Figure 1: A bird’s eye view of the occluding wall and a possible laser spot position. Here, the laser spot is at an angle χ measured around the edge into the visible scene. At this angle, the small wedge shown in red is not illuminated by the laser and thus not recoverable by our algorithm. The alternative position illuminates more of the hidden scene but with more signal attenuation for certain parts of the hidden scene due to greater distances between bounces.

(but not necessarily) with an example of how things degrade when the edge is not within the line of sight of the camera+laser?

Response:

Though there are several issues to address here, we kept this block together as presented by the reviewer. First, we feel that “Either way, also looking at the SM, the camera in reality does always have direct line of sight access to most of the room and definitely LOS access to the objects” reflects a misunderstanding. We realize that the original location of the laser spot in Figure 1A and Figure 4A of the main paper may have been misleading. In response to your comment, we have **updated both figures** so that the laser spot is clearly on the visible side of the wall. What is and is not visible by direct line-of-sight (LOS) is illustrated in Figure 1A of the main paper. Conceptually, LOS view ends at the extension of the wall with the exploited vertical edge, which we consider to be the boundary between the visible and hidden parts of the scene. (It may help to imagine that the imaging equipment in Figure 1A has zero volume and sits flush with the wall. Then the extension of the wall more plainly coincides with the limit of LOS view from the perspective of that imaging equipment. Substantially, this simplification describes the configuration of interest.) As shown in Figure 1A of the main paper, our setup requires (1) that the SPAD FOV be pointed to a region of the floor that is adjacent to the vertical edge; and (2) that the laser spot is directed at the floor in order to illuminate a large portion of the hidden scene. The sensor and laser themselves can be situated in a variety of positions on the visible side of the wall with no LOS view of the hidden scene, as long as they can be directed towards the aforementioned positions. In the same way that other NLOS imaging techniques choose which portions of the relay surface to illuminate and observe, we select which part of the floor (our relay surface) to illuminate and observe.

The reviewer raises an excellent question about whether, “the quality of the reconstructions decay rapidly as soon as the laser spot and camera field of view are shifted away from the edge of the wall”. In the next few paragraphs, we will argue that the laser spot position is actually quite flexible, while the reviewer’s intuition about the camera FOV position is absolutely correct. These are addressed by the paragraph inserted in the main paper starting at line 269.

Our method does *not* require the laser spot to be directed to the hidden side of the occluding wall to perform NLOS imaging. In fact, the laser spot may even be shifted into the visible side, as shown from a bird’s eye view in Figure 1. (This is also newly added as Figure 8 of the Supplementary Notes.) Here, the laser spot is at angle χ measured around the

Figure 2: In (A), we explore the merits of three different FOV locations: red, green, and blue. Red, green, and blue uncertainty bubbles above depict estimate uncertainty at different locations in the hidden space. The footprint of the occluding wall is marked with a thick black line and has a vertical edge at the origin. In (B) we plot an image of histogram max at each pixel, for each FOV, for a target located in range at 1 m and in angle at $7\pi/12$. For this particular target location, the red and green FOVs have a prominent ‘penumbra’ shadow cast across them due to the occluding edge; the blue FOV is farther away from the edge and thus only exhibits a subtle radial falloff pattern. We note that when the FOV contains a ‘penumbra’ pattern, uncertainty in angle becomes very small. FOVs that are farther from the edge, like the blue FOV, are less likely to contain useful penumbra pattern. Thus, as the reviewer suggested, FOVs that are farther from the edge pose a more challenging reconstruction problem.

edge into the visible side. In this position, although a small wedge of the hidden scene (red) is not illuminated by the pulsed laser and is thus not recoverable, the remaining majority of the hidden volume is recoverable. In order to image more of the hidden scene, the laser spot might be directed towards the “Alternative position” marked in Figure 1. While this laser spot illuminates more of the hidden scene, the laser spot is farther from the laser itself and certain parts of the hidden scene, resulting in lower signal power. Thus, as long as the laser has an unobstructed view of the floor on the visible side, a laser spot may be chosen to balance the imaging needs of a given application.

With regard to the SPAD FOV location, our reconstruction method relies heavily on the occlusion provided by the vertical edge; this is the aspect that is inspired by the corner camera [4]. For example, we use processing similar to the passive corner camera to count the number of hidden targets and to initialize target parameters; see Figure 6 of the Supplementary Notes. For this reason, we do not expect our algorithm to work well when the SPAD FOV is farther from the corner.

Results of a Cramer–Rao bound (CRB) analysis included in the first author’s PhD dissertation [6, Sect. 5.5] are shown in Figure 2 and provide more general (algorithm-blind) insight about different SPAD FOV positions. In this study, the CRB was derived for estimating a small hidden facet’s unknown parameters using measurements from the active corner camera.

Note that this simple toy scenario is different from the more complicated and challenging experimental scenarios we have demonstrated in this paper; the goal was to use this simple scenario to explore *qualitative* trends. Because the analysis is for localization of a small facet target, we did not add this material to the Supplementary Notes. Instead, the first author’s PhD dissertation is referenced in the added paragraph that starts at line 269.

Figure 2(A) depicts theoretical spatial uncertainty regions for a hidden target for the red, green, and blue FOV positions. The color of the uncertainty region matches the corresponding FOV position. The dark black line shows the footprint of the occluding wall. Note that the occluding edge is located at the origin. These uncertainty regions – within which the majority of a target’s estimates are expected to fall – were computed using the CRBs derived in [6, Sect. 5.5]. Figure 2(B) depicts the histogram maximum, in units of photon counts, for the three different FOVs when the hidden facet is located in angle at $\frac{7}{12}\pi$. Note that for this particular target location, the red and green FOVs have a prominent ‘penumbra’ shadow cast across them due to the occluding edge; the blue FOV is farther away from the edge and thus only exhibits a subtle radial falloff pattern. We observe that angular uncertainty becomes very small when an FOV contains a ‘penumbra’ shadow, and it is much larger when it does not (as is often the case with the blue FOV and other locations farther from the occluding edge). Thus, as the reviewer suggested, we expect FOVs that are farther from the occluding edge to pose more challenging reconstruction problems.

In general, the reviewer’s questions about the position of the laser spot and SPAD field of view made us realize that our experimental design choices could be described with more clarity. To address this, we have added a **substantially new Supplementary Note 3** where we explain how these design choices, made to either emulate more modern hardware or simplify our experiment, do not change the fundamental concepts that have been demonstrated. In particular, Section 3.2 in the Supplementary Notes document details the procedure followed in the lab to emulate the hard-gating ability of newer SPAD arrays. (The previous Supplementary Note 3 is now Section 3.1 of a much longer Supplementary Note 3.)

3. My impression from reading the main text was that motion was needed in order to reconstruct the position and shape of the object. But after reading the SM, if I understand correctly, motion of the object is only required in order to map the static background whereas the object itself can be (and is) reconstructed from each individual frame, i.e. in principle motion is not needed? I do not see anything in the model and maths that indicates that motion is needed for the object reconstruction, i.e. this could be done from just one single frame. Could the authors comment on this and clarify in the text?

Response:

Thank you for drawing our attention to this possible source of misunderstanding. You are correct that we process each measurement frame separately, thus not relying on frame-to-frame motion. (This is a virtue for the amount of data to be handled at once and for parallelizability.) We combine estimates of ψ_{oc} occluded background parameters across frames to produce the background map.

We have attempted to clarify with a sentence inserted starting at line 136,

“Conceptually, this measurement of \mathbf{b} allows us to subtract the counts due to an arbitrary and unknown stationary hidden scene environment; the actual computations are more statistically sound than simple subtraction.”

and by inserting “each processed separately,” at line 175.

Response to Reviewer #2

1. “Non-line-of-sight tracking and mapping with an active corner camera” present new results on the NLOS imaging with an edge-resolved transient imaging (ERTI) device. This method records transient information from an inaccessible area around a corner by illuminating the hidden scene with laser light scattered from a surface (e.g. floor). With the help of an edge (e.g. door frame) in the scenery different areas are sampled. During the evaluation, a sector-by-sector analysis can be performed by difference formation.

The manuscript is well written, theory and experimental results are presented in detail and show an amazing quality.

However, in another publication (Nature Communications, 2020), parts of the author team have presented a very similar procedure.

Despite the fact that a slightly different type of approach was used, the procedures, scenarios and results are very similar that it is difficult to distinguish between them, giving the impression of marginal innovation.

In the present paper, I would have liked to have seen somewhat clearer use cases investigated and the system used outside of a controlled lab environment with less than ideal walls (smooth white surfaces). Furthermore, I am surprised that it is not possible to achieve a better reconstruction than the one already presented in 2020.

Although I really enjoyed reading this manuscript and there are no complaints in terms of content and form, I am afraid that the presented procedure and results do not represent the necessary innovation required for publication in such a prestigious journal as Nature Communications. Therefore, I cannot recommend the publication of the manuscript. Maybe, it would be worth to submit this paper to Scientific Reports instead.

Response:

Though we are naturally disappointed by your overall judgment that our work should not be published in *Nature Communications*, we appreciate your praise on the writing and “amazing quality” of results. Whether or not convincing you to change your recommendation is required for publication, we would like to convince you. Much of the opening section of this document is meant to address your misgivings. We will attempt to address your questions without being excessively repetitive.

We believe that the conceptual and technical innovations here are worthy of wide dissemination. The technical innovation of having efficient computations without a confocal approximation is essential to undertaking the use of an array detector. With that array detector, we can achieve snapshot-style reconstruction that is not limited to a point-like hidden scene model. Furthermore, our explicit occluded background modeling and recovery is a new capability.

We certainly do not deny that there are some conceptual commonalities between the present work and ERTI [2]. Table 1 and its caption summarize major differences and *non-conceptual* reasons for the foreground resolution of the present work to not improve upon ERTI.

With regard to expanding the experimental results, that is certainly always on the wish list. Space and approval for laser-based experimentation outside of a laboratory are not easy to obtain. We did consider a variety of test scenarios including moving objects and presence of ambient light, which were not included in previous work. The use of white or retroreflective objects and walls has been prevalent in proof-of-concept experiments in the literature [7–10]. We were pleased to have good results with a fabric-surfaced mannequin that is certainly non-planar.

Table 1: Summary of differences between the present work and ERTI [2]. A method for fast approximate computation of facet response without confocal approximation enables arbitrary illumination position. The method works in a snapshot mode (no scanning of illumination or detection), and by combining information across multiple frames we can form a reconstruction of the occluded background. However, foreground reconstruction resolution is limited by very coarse timing resolution of the available hardware. Note also the data transfer limitations detailed in Section 3.1 of the Supplementary Notes.

	ERTI [2]	This Work
Illumination		
position	at vertical edge for confocal approximation	arbitrary
number of positions	45 spots along a small arc	single position
Measurement		
detector	single-element SPAD	32 × 32 pixel SPAD array (snapshot)
time bin	16 ps	390 ps
Reconstruction capability		
foreground?	yes	yes
occluded background?	no	yes

References

- [1] D. Shin, F. Xu, D. Venkatraman, R. Lussana, F. Villa, F. Zappa, V. K. Goyal, F. N. C. Wong, and J. H. Shapiro, “Photon-efficient imaging with a single-photon camera,” *Nat. Commun.*, vol. 7, 24 Jun. 2016.
- [2] J. Rapp, C. Saunders, J. Tachella, J. Murray-Bruce, Y. Altmann, J.-Y. Tournet, S. McLaughlin, R. M. A. Dawson, F. N. C. Wong, and V. K. Goyal, “Seeing around corners with edge-resolved transient imaging,” *Nat. Commun.*, vol. 11, no. 5929, Nov. 2020.
- [3] W. Krska, S. W. Seidel, C. Saunders, R. Czajkowski, C. Yu, J. Murray-Bruce, and V. K. Goyal, “Double your corners, double your fun: The doorway camera,” in *Proc. IEEE Int. Conf. Comput. Photography*, Pasadena, CA, Aug. 2022.
- [4] K. L. Bouman, V. Ye, A. B. Yedidia, F. Durand, G. W. Wornell, A. Torralba, and W. T. Freeman, “Turning corners into cameras: Principles and methods,” in *Proc. 23rd IEEE Int. Conf. Computer Vision*, 2017, pp. 2270–2278.
- [5] C. A. Metzler, D. B. Lindell, and G. Wetzstein, “Keyhole imaging: Non-line-of-sight imaging and tracking of moving objects along a single optical path,” *IEEE Trans. Comput. Imaging*, vol. 7, 2021.
- [6] S. W. Seidel, “Edge-resolved non-line-of-sight imaging,” Ph.D. dissertation, Boston University, Sep. 2022.
- [7] A. K. Pediredla, M. Buttafava, A. Tosi, O. Cossairt, and A. Veeraraghavan, “Reconstructing rooms using photon echoes: A plane based model and reconstruction algorithm for looking around the corner,” in *Proc. IEEE Int. Conf. Comput. Photography*, 2017, pp. 1–12.
- [8] F. Heide, M. O’Toole, K. Zang, D. B. Lindell, S. Diamond, and G. Wetzstein, “Non-line-of-sight imaging with partial occluders and surface normals,” *ACM Trans. Graph.*, vol. 38, no. 3, pp. 1–10, May 2019.
- [9] M. O’Toole, D. B. Lindell, and G. Wetzstein, “Confocal non-line-of-sight imaging based on the light-cone transform,” *Nature*, vol. 555, pp. 338–341, Mar. 2018.
- [10] D. B. Lindell, G. Wetzstein, and M. O’Toole, “Wave-based non-line-of-sight imaging using fast f - k migration,” *ACM Trans. Graph.*, vol. 38, no. 4, pp. 116:1–116:13, Jul. 2019.

REVIEWERS' COMMENTS

Reviewer #1 (Remarks to the Author):

The authors have replied to all of my questions and I find the manuscript to now be significantly clearer and improved.

Congratulations are due to the two lead authors who have moved on to new jobs. But aside from this, the authors have pointed out the challenging context of a lab move etc.

To be fair though, I do not think that any of these factors have negatively impacted the quality of the reply and of the final manuscript.

I also agree with the authors' point (which was also the starting sentence in my first review) that 'new and old' techniques have been brought together in this work and, as can be the case, the end result is more than the sum of the parts and required innovation in the algorithms and how the hardware is used to finally obtain results that I do not believe have been shown before and that advance the state of the art of NLOS imaging.

Overall, this is very high quality work and makes a significant contribution to a field that is of interest to a broad range of researchers and relevant to a broad range of application scenarios.

I would therefore confirm my original recommendation and I would suggest acceptance without the need for any further modifications.

Reviewer #2 (Remarks to the Author):

I would like to thank the authors for constructively taking my sometimes somewhat harsh suggestions and significantly improving the manuscript so that it is now at a level that can be considered for publication in Nature Communication.

The manuscript, based on an approach from an earlier publication, describes a method that uses a combination of corner camera and time-of-flight measurements to capture areas beyond the direct line of sight. Unlike an earlier approach, a array sensor is used instead of a scanning single pixel. Although this development step is obvious, it is significant because it makes the method interesting for more application areas. Besides the faster acquisition of data sets, parallel acquisition, the observation of non-static scenarios is especially noteworthy at this point.

The manuscript is well structured and on an excellent language level. The results are convincing and of interest to a broader audience.

With the contributed changes, the manuscript is now at a level that can be recommended for publication in Nature Communication.